# ProInfer: An interpretable protein inference tool leveraging on biological networks

Hui Peng[1,2], Limsoon Wong[3]*, Wilson Wen Bin Goh[1,2,4]*

**1** Lee Kong Chian School of Medicine, Nanyang Technological University, Singapore, Singapore, **2** School of Biological Sciences, Nanyang Technological University, Singapore, Singapore, **3** Department of Computer Science, National University of Singapore, Singapore, Singapore, **4** Center for Biomedical Informatics, Nanyang Technological University, Singapore, Singapore

* wongls@comp.nus.edu.sg (LW); wilsongoh@ntu.edu.sg (WWBG)

## Abstract

In mass spectrometry (MS)-based proteomics, protein inference from identified peptides (protein fragments) is a critical step. We present ProInfer (Protein Inference), a novel protein assembly method that takes advantage of information in biological networks. ProInfer assists recovery of proteins supported only by ambiguous peptides (a peptide which maps to more than one candidate protein) and enhances the statistical confidence for proteins supported by both unique and ambiguous peptides. Consequently, ProInfer rescues weakly supported proteins thereby improving proteome coverage. Evaluated across THP1 cell line, lung cancer and RAW267.4 datasets, ProInfer always infers the most numbers of true positives, in comparison to mainstream protein inference tools Fido, EPIFANY and PIA. ProInfer is also adept at retrieving differentially expressed proteins, signifying its usefulness for functional analysis and phenotype profiling. Source codes of ProInfer are available at https://github.com/PennHui2016/ProInfer.

## Author summary

Protein inference is a key step in proteomics data analysis. However, this procedure suffers from coverage issues due to high statistical stringency requirement and noise. Integration of prior knowledge to guide protein assembly can be a powerful approach. Hence, we developed a novel protein inference tool ProInfer that incorporates a length-adjusted and weighted-accumulated posterior error probability score with protein-complex networks. Compared against existing tools, ProInfer achieves the highest recall and F1 score in protein inference and also identifies novel differentially expressed proteins not reported by any other tool.

This is a *PLOS Computational Biology* Methods paper.

**Data Availability Statement:** All relevant data are within the manuscript and its Supporting Information files.

**Funding:** This work was supported by the Ministry of Education Singapore via an AcRF Tier 2 award

(MOE2019-T2-1-042 to WWBG and LW) and a AcRF Tier 1 award RT11/21 to WWBG). The funders had no role in study design, data collection and analysis, decision to publish, or preparation of the manuscript.

**Competing interests:** The authors have declared that no competing interests exist.

## Introduction

Contemporary mass spectrometry (MS)-based proteomics is characterized by advanced high-throughput technologies for identifying proteins from complex mixtures [1–2]. MS proteomics measures the mass-to-charge (m/z) ion ratios, retention times, and ion intensities of precursor ions and peptide fragments [3–4]. A complex multi-step analytical procedure is then performed to reverse-engineer spectral information into peptide sequences (peptide-spectrum matching), followed by re-assembly of peptides to constituent proteins. The process of estimating the optimal set of proteins given acquired spectra is known as the protein inference problem [5–7]. Quantitative analysis is then performed to identify phenotype-specific proteins [8], obtain their function annotation [9] and determine potential applications in clinical [10] and medical settings [11] (**Fig 1A**).

In peptide-spectrum matching, a spectrum is matched against peptides in both a reference and a decoy protein sequence database, producing a score for each peptide-spectrum match (PSM) [12]. Given these PSMs, one can perform peptide identification, i.e., distinguishing correct PSM from incorrect ones, with well-known tools such as PeptideProphet [13] and Percolator [14]. Percolator was shown to identify more PSMs under similar q-value thresholds [14]. It can output q-values and posterior error probability scores (PEP) for identified peptides. Briefly, a q-value is the minimal false discovery rate (FDR) threshold needed for a positive identification of a given peptide while a PEP score indicates the probability of incorrectly identifying a non-existing peptide [15]. PEP scores are reported in popular protein inference tools such as Fido [16], PIA (Protein Inference Algorithms) [17] and EPIFANY [18].

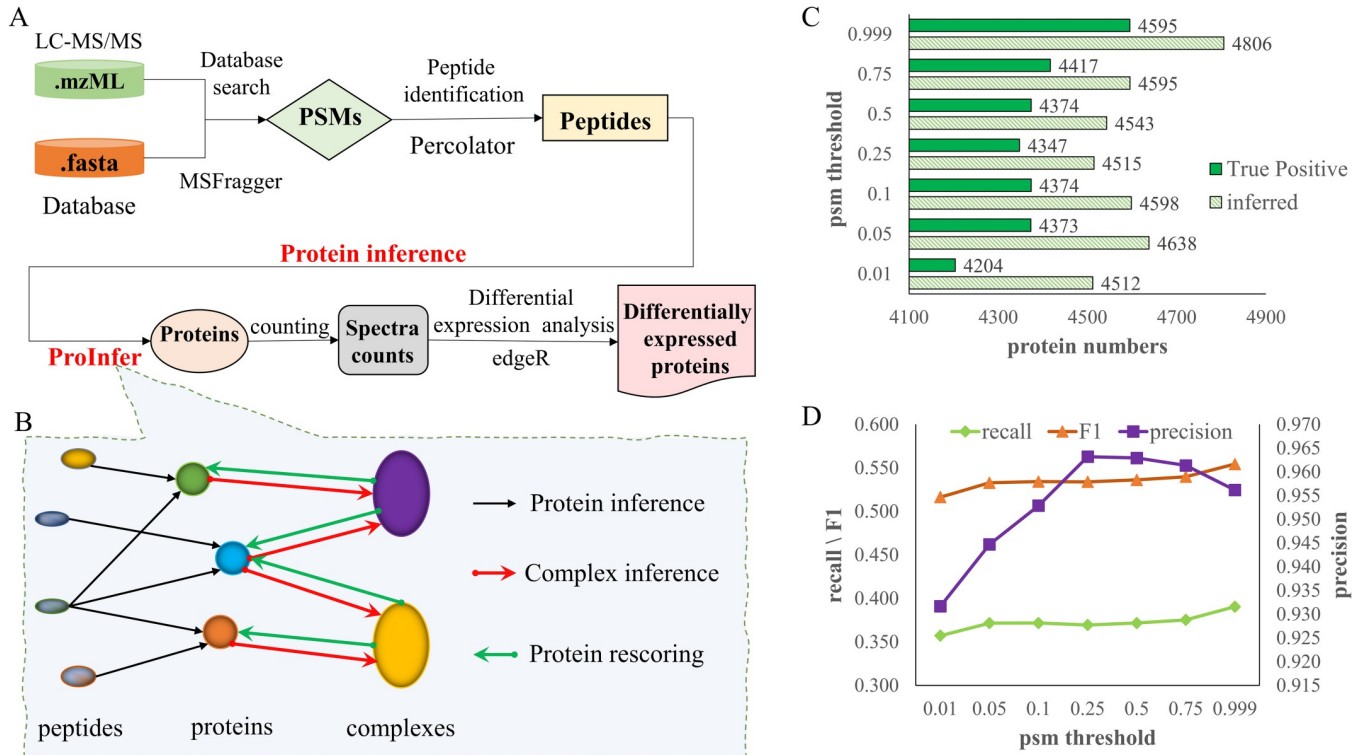

**Fig 1. Workflow for protein inference and differential expression analysis with proteomic data and hyperparameter optimization results of proposed ProInfer.** A shows a simple workflow of proteomics data-based protein inference and differential expression analysis. B gives a schema diagram of how ProInfer works. C presents the average inferred protein numbers and numbers of true positives found by ProInfer under different psm filtering thresholds (y-axis). The curves in D illustrate the performances of ProInfer indicated by recall, F1 (left-y axis) and precision (right-y axis) affected by various psm filtering thresholds (x-axis).

Protein inference is concerned with protein identification from identified peptides. Protein inference is a demanding task and is affected by experimental/biological and analytical challenges. Example experimental/biological challenges include incomplete proteome coverage due to high dynamic range of protein abundances, limitations in digestion and protein separability under experimental running conditions, and detector sensitivity and resolution [18], whereas the algorithm design and assumption validity, parameter values, and sequence library completeness are example analytical challenges. A particularly cumbersome problem is dealing with peptide ambiguity where one peptide can be mapped to two or more proteins [19]. Proteome coverage issues can be eased by leveraging on alternative data acquisition strategies, e.g., parallel accumulation–serial fragmentation combined with data-independent acquisition (dia-PASEF) [20], which increases precursor identification specificity. Peptide ambiguity problem is solved by either discarding ambiguous peptides (peptides which map to > 1 protein), e.g., Percolator [14]; or conducting network analysis on peptide-protein bipartite networks, e.g., EPIFANY [18]. Those proteins sharing the same constituent peptides may be reported as protein groups [17] in which case, we can be assured that at least 1 member in the protein group exists. Protein inference is a critical problem for proteomics, and so, many such methods have been developed (please see Huang et al [21] for details regarding some early methods). Newer (and popular) protein inference methods include Percolator [14], Fido [16], PIA [17], EPIFANY[18] and ProteinProphet [22].

Percolator [14] was originally designed for post-processing of peptide-spectrum matching results using semi-supervised learning. When protein inference is required, users may opt for a protein-level FDR threshold to output inferred proteins and their respective probabilities [23]. Perhaps to improve precision, Percolator does not consider ambiguous peptides, which may reduce proteome coverage. Fido [16] is a Bayesian probabilistic method for addressing ambiguous peptide problems and computing protein probabilities using graph-transforming algorithms. Fido creators claimed their method outperforms the heuristic posterior probability models based on expectation-maximization such as ProteinProphet [22]. PIA [17] is a consensus tool for integrating results of different search engines and different protein inference tools, e.g., ProteinProphet [22], Scaffold [19], and IDPicker [24]. PIA addresses ambiguous peptides by employing maximum parsimony principles and finding a minimal set of proteins explaining found peptides or PSMs [18]. EPIFANY [18] is a recently published protein inference tool that applies a loopy belief propagation algorithm (LBP) with convolution trees to process Bayesian networks. Via a peptide-protein bipartite graph, EPIFANY adopts convolution trees to propagate probabilities between peptides and proteins even for ambiguous peptides.

While these popular tools play significant roles in protein inference, there is room for improvement, especially in reported protein accuracy and proteome coverage. Current protein inference methods generally report proteins with lower q-values (q-values in this scenario, is a rank-based metric for comparing confidences of inferred proteins being present [15]). Consequently, proteins that are in fact present but have lower peptide support are ignored. Unless approaches exist to exhaustively mine low quality spectra for peptide spectra matches (PSMs), we do not expect peptide information supporting each protein to change drastically. Given such constraints, we believe most current tools are reliant only on direct peptide-to-protein information, to attain an upper limit (albeit incomplete) on the observable proteome [21]. To overcome these information barriers, we believe incorporating prior knowledge via data integration, e.g., drawing from independent data sources such as biological networks, is essential. Hence, we propose a protein-length adjusted posterior error probability accumulation method *ProInfer* (short for Protein Inference), which features a comprehensive yet simple rule towards protein scoring (including how it handles peptide ambiguity). To help users, ProInfer's methodology is easily understood; involving no complex calculations while possessing reasonable

assumptions. Additionally, ProInfer borrows similar principles from our missing protein prediction method PROTREC [25] that leverages on the phenomenon that proteins forming a stable protein complex (or constituting part of a tightly clustered network module) are more likely to be co-expressed [26]. Specifically, it incorporates protein complex information to rescue proteins with weak signals themselves but have neighbors with strong evidence. ProInfer achieves excellent performance in both protein inference and downstream quantitative analysis.

## Materials and methods

### ProInfer

A schematic of ProInfer is shown in **Fig 1B**. ProInfer comprises three stages: protein inference from the peptide-protein network, protein complex inference and protein rescoring from the protein-complex network. To define these terms: A peptide-protein network refers to the connections of identified peptides to their host proteins. This information is vital to demonstrating existence of these proteins in the sample (see an example of a peptide-protein network in left side of **Fig 1B**). Peptide-protein networks are useful for performing protein assembly from constituent peptides. Proteins work together as aggregates known as biological networks. These in turn, can be expressed as protein complexes or pathways, and is information rich. Thus, a protein-complex network (see the right-side bidirectional network in **Fig 1B**) is composed of proteins aggregating into protein complexes (defined as groups of polypeptide chains linked by noncovalent protein-protein interactions [27]). An arrow from a protein pointing towards a protein complex denotes the protein belongs to that complex. And thus, the protein's existence adds evidence supporting the presence of the complex in a tissue (i.e., protein complex inference process based on observed constituent proteins). Alternatively, an arrow from a protein complex to a protein propagates the existence probability of this complex to its constituent proteins, helping to re-evaluate our confidence in the protein's existence in a tissue (i.e., protein rescoring).

### Protein inference from peptide-protein network

Our idea for protein inference from peptide-protein network stems originally from a need to address peptide ambiguity issues. Suppose an identified peptide $Q$ from spectra is mappable to $N$ proteins $\{P_1, P_2, . . ., P_N\}$. We may reasonably assume that Q has an equal chance to come from $N$ candidate proteins.

The false-reporting probability of using Q to support reporting $P_1$ is the chance that $Q$ is an incorrect identification (with PEP score of $pep_Q$) plus the chance that $Q$ is not incorrect but does not come from $P_1$ (i.e., it is from one of $P_2, . . ., P_N$). This may be expressed as:

$$pep_Q + (1 - pep_Q) \cdot (N - 1)/N \tag{1}$$

where $pep_Q$ is the PEP score of peptide Q and N is the number of mappable proteins.

Eq 1 can be rewritten as

$$1 - (1 - pep_Q) \cdot (1/N) \tag{2}$$

Here, the term $(1 - pep_Q) \cdot (1/N)$ means each candidate protein receives equal support, i.e., $Q$'s posterior probability $(1 - pep_Q)$. In practice, $Q$ should not arise equally from each candidate parent protein as "longer proteins are more likely to generate spurious matches than shorter ones" [28]. In other words, the support attributed by an ambiguous peptide towards the existence of a protein also depends on the protein's length. Thus, to express this idea, we normalize

the probability of $Q$ from $P_1$ by accounting for protein length. The error of using $Q$ to support $P_1$ can be modified to

$$1 - (1 - pep_Q) \cdot \phi \tag{3}$$

where $\phi$ is a length-based adjustment and is computed as:

$$\phi = \frac{\sum_{i=1}^{N} len(P_i)/(len(P_1))}{\sum_{h=1}^{N}(\sum_{i=1}^{N} len(P_i)/(len(P_h)))} \tag{4}$$

where $len(P_i)$ is a function computing the length of protein $P_i$.

Now we consider the problem of reporting a protein $P_1$ to be present given supporting peptides $\{Q_1, Q_2,\ldots Q_j,\ldots,Q_m\}$. If a peptide $Q_j$ supports a protein $P_1$ uniquely, this peptide is considered "unique". If a peptide $Q_j$ supports n > 1 proteins $\{P_1, P_2,\ldots P_i,\ldots,P_n\}$, this peptide is considered "ambiguous".

The error of peptide $Q_j$ supporting $P_1$ can be context-driven (unique or ambiguous) as follows:

$$p(P_1 \text{ is a false report}|Q_j) = \begin{cases} pep_{Q_j}, & Q_j \text{ is a unique peptide} \\ 1 - (1 - pep_{Q_j}) \cdot \phi, & Q_j \text{ is a ambiguous peptide} \end{cases} \tag{5}$$

where $\phi$ is computed as shown in Eq (4).

We assume that peptide $Q_1$ supporting $P_1$ is independent of other peptides $Q_{j,j\neq1}$ supporting $P_1$. The total errors of using $\{Q_1, Q_2,\ldots Q_j,\ldots,Q_m\}$ to support $P_1$ to be present can be computed by:

$$p(P_1 \text{ is a false report }|\{Q_1, Q_2, \ldots Q_j, \ldots, Q_m\}) = \prod_{j=1}^{m} p(P_1 \text{ is a false report}|Q_j) \tag{6}$$

To calculate the FDR, we transform the cumulative PEP score from (6) (denoted as *accPEP*) to a confidence score:

$$S = -10 \cdot \log_{10}(accPEP + 1e - 14) \tag{7}$$

In (7), we spike a small value of 1e−14 to avoid errors where $S$ becomes undefined (NaN) when accPEP is 0. We compute the FDR derived q-value for reported proteins in the same way as EPIFANY [18]. Firstly, the reported $L$ proteins are ranked by confidence scores ($S$) in descending order, i.e., $\{P^{r1}, P^{r2},\ldots,P^{rk},\ldots,P^{rL}\}$. Then, the q-value of $P^{rL}$ is the FDR with the threshold of its confidence score calculated by

$$qvalue(P^{rL}) = FDR(x = S(P^{rL})) = \frac{|\{y \geq x, y \in D\}| + 1}{|\{y \geq x, y \in T\}| + 1} \tag{8}$$

where $x$ is the threshold, and $|\{y{\geq}x, y{\in}D\}|$ or $|\{y{\geq}x, y{\in}T\}|$ counts the number of decoy (D) or target proteins (T) with confidence scores no less than the threshold.

For $k \in[L{-}1, 1]$,

$$qvalue(P^{rk}) = min\{FDR(S(P^{rk})), qvalue(P^{r(k+1)})\} \tag{9}$$

Given these q-values, we can select an appropriate FDR, e.g., 1%, to report the proteins that qualify under this threshold. At FDR 1%, we expect 1 decoy protein (False Positive) per 100 correct target proteins (True Positive).

## Protein complex inference and protein rescoring with protein-complex network

Given only PSM information, we may easily reach the upper boundary of reportable proteins no matter how good the protein inference tool is. This is because these protein inference methods are ultimately dependent on spectra completeness and quality [21]. In a typical experiment setting, due to limitations in instrument sensitivity, protein abundance and protein sequence uniqueness, some proteins are only supported by weak signals (e.g., few supporting peptides and/or low confidence peptides), and thus, are difficult to observe. To rescue such proteins, and improve proteome coverage, we may "borrow" information from other important modalities (protein-protein interaction network [29], gene expression profiles [30], etc.) Biological network information encapsulated in the form of protein complexes is particularly valuable, possessing high biological information value [31], improving statistical reproducibility [32] and improving phenotype characterization [33]. Using protein complexes, we developed PRO-TREC [25], a tool for missing protein recovery, which outperforms other missing protein prediction methods. We hypothesize that protein complexes can also be useful for improving protein inference from peptide information. To test this idea, we use protein complex information in ProInfer.

Suppose ProInfer (refers to the part described in the above section) outputs $L$ candidate proteins and their accPEP scores and q-values denoted by:

$$Pros = \{(P_i, accPEP_i, q_i) | i \in [1, L]\} \tag{10}$$

We collected $C$ reliable protein complexes and generated $C$ decoy protein complexes by replacing the protein ids (e.g., sp|P41182) in each real complex with corresponding decoy protein ids (e.g., DECOY_sp|P41182). The complexes are denoted by:

$$Cpxs = \{(c_j, cP^j) | j \in [1, 2C]\} \tag{11}$$

where $c_j$ is the $j$th known protein complex, and it contains $X$ constituent proteins denoted by $cP^j = \{cP_1^j, cP_2^j, \ldots, cP_X^j\}$.

The following procedures describe the protein complex inference and integration of protein complex information in $Cpxs$ with ProInfer's outputs $Pros$:

**Step1.** Initialization. We initialize the probability of protein $P_i$ being present in the sample as $p(Pi) = 1 - accPEP_i$. For a given FDR $f$, ProInfer reports $num0$ numbers of target proteins.

**Step2.** Protein complex inference. Calculate the probability of a protein complex $c_j$ being present in the sample as the maximum probability of the subset proteins in this complex and in $Pros$, denoted by:

$$p(c_j) = \max\{1 - accPEP_a | a \in [1, z]\} \tag{12}$$

where $accPEP_a$ is the accPEP score of the $a$th protein in the subset $cP^j \cap Pros$.

**Step3.** Calculate the probability of protein $P_i$ in $Pros$ being present in the sample according to protein complex information. Let $\{c_1, c_2, \ldots, c_Q\}$ be the $Q$ complexes containing $P_i$, then the probability of $P_i$ is computed as the maximum probability of $\{c_1, c_2, \ldots, c_Q\}$:

$$p\_cpx(P_i) = \max(p(c_1), p(c_2), \ldots, p(c_Q)) \tag{13}$$

If no protein complex contains $P_i$, then $p\_cpx(P_i) = 0$.

**Step4.** Update the probability of $P_i$ being present in the sample. By comparing $p(P_i)$ with $p\_cpx(P_i)$, we update the probability of $P_i$ being present in the sample as:

$$p(P_i)' = \max(p(P_i), p\_cpx(P_i)) \tag{14}$$

**Step5.** Check whether we can now report new target proteins with given FDR *f*. From Step4, we get $accPEP_i' = 1 - p(P_i)'$. We transform $accPEP'$ to its confidence score via above Formula (7). Then, via Formulas (8) and (9), we compute the q-value of $P_i$ as $q_i'$. *Pros* is updated as following formula:

$$Pros = \{(P_i, accPEP_i', q_i')|i \in [1, L]\} \tag{15}$$

With FDR *f*, *num* target proteins are reported. If *num > num0*, then turn to **Step1**, otherwise output *Pros* and stop.

Unlike PROTREC, ProInfer does not compute the probability of a protein complex being present in a biological sample based on the weighted probability of all observed constituent proteins [25]. ProInfer's approach is based on calculating the maximum of constituent proteins posterior probability (PP) expressed as PP = 1—*accPEP*. We used protein complexes downloaded from CORUM 3.0 [34]. We compute the probability of a protein complex being present in the sample as the maximum probability of its proteins' posterior probability measured by 1—*accPEP*. Then, we update the posterior probability of a protein being present with the higher value compared between the protein's original PP value and its parent complexes' posterior probabilities, i.e., max(original PP, complexes' PPs). For decoy proteins, PP derived from corresponding decoy complexes will be used (similarly, calculated by max(original PP, complexes' PPs)). A decoy complex is constructed by replacing target proteins in its twin true complex with decoy ones. The introduction of decoy complexes is to make the inference of both target proteins and decoy proteins in a similar way to avoid bias in estimating FDR. This propagation procedure is iterated until no additional target proteins are reported under a given FDR.

## Hyperparameter optimization and datasets

Protein inference is conducted following peptide identification, where PSMs are evaluated and then filtered by a given PEP threshold. Retained PSMs are regarded as reliable. A strict PEP threshold retains high confidence PSMs but also produces many false negatives. Conversely, relaxed PEP thresholds alleviate the false negative problem but at the cost of introducing more false positives. Different tools adopt different strategies for threshold optimization.

Tools such as EPIFANY, Fido and PIA have some tool-specific hyperparameters to be tuned, e.g., greedy group resolution for EPIFANY [18] and Fido [16], regularization type for EPIFANY [18], and input score type and scoring method for PIA [17]. For a fair comparison, we optimized their hyperparameters with the same Hela cell line dataset initially. The Hela cell line, derived from cervical cancer cells, is the oldest and most used human cell line [35]; it is well-documented and widely applied in biochemical, biological, and medical experiments [36]. 4-replicates Hela cell line raw data of Mehta et al [37] were downloaded from PRIDE [38] with Project ID PXD022448 (see **Table 1**).

For performance benchmarking, the lung cancer data (lung cancer) of Li et al [39] (PXD000853), the THP1 cell line and RAW264.7 mouse macrophage cell line of Li et al [40] (PXD019800) were used. THP1 is a human leukemia monocytic cell line and is commonly studied for estimating modulation of monocyte and macrophage activities [41]. RAW264.7 is a mouse leukemia cell line of monocyte macrophage, where it has been extensively used to

**Table 1. Summary of datasets used for hyperparameter optimization and performance evaluation.**

| Dataset | Condition | Replicates/Samples | PRIDE ID | Purpose |
|---|---|---|---|---|
| Hela | - | DDA1,DDA2,DDA3,DDA3 | PXD022448 | hyperparameter optimization |
| THP1 | M0 | M0_1, M0_2, M0_3 | PXD019800 | Performance benchmarking |
| THP1 | M1 | M1_1, M1_2, M1_3 | PXD019800 | Performance benchmarking |
| RAW264.7 | M0 | M0_1, M0_2, M0_3 | PXD019800 | Performance benchmarking |
| RAW264.7 | M1 | M1_1, M1_2, M1_3 | PXD019800 | Performance benchmarking |
| lung cancer | Normal | N24742,N31945,N32813_r,N35480 | PXD000853 | Performance benchmarking |
| lung cancer | Patient | T24742,T31945,T32813_r,T35480 | PXD000853 | Performance benchmarking |

study macrophage functions, mechanisms, and signaling pathways [37,42]. The lung cancer data was adopted to discover new anticancer therapeutic targets [39] (see **Table 1**).

## Proteomic dataset processing

Raw data were converted to.mzML format with MSConvert [43] and processed as per flow-chart in **Fig 1A**. MSFragger-3.4 [44] was used to conduct database search. A target-decoy searching strategy [12] was adopted where the protein database contains human reviewed proteins from UniProt [45] (UP000005640, downloaded in 5/5/2022) and known contaminants from the common Repository of Adventitious Proteins (cRAP, https://www.thegpm.org/crap/, added by FragPipe-17.1 [44], https://fragpipe.nesvilab.org/) database together with the decoy proteins generated by sequence reversal. Search parameters are as follows: precursor mass tolerance (PMT) of 20ppm, fragment mass tolerance (FMT) of 0.05Da, and peptide length of 7 to 50 (remaining parameters are left as default). Prior to inputting to different protein inference tools, we performed peptide indexing and feature extraction with OpenMS (version 2.7.0) [46] for Percolator, which was then used to conduct peptide identification with PEP scores computed for each PSM. With these scored PSMs, hyperparameters were optimized with grid search: PSM filtering thresholds (PSMs with PEP scores bigger than the threshold are dropped) were ranged among [0.01, 0.05, 0.1, 0.25, 0.5, 0.75, 0.999], while for other tool-specific hyperparameters, all possible values are tested.

## Validation and performance evaluation

The protein-expression tissue database, Human Protein Atlas (HPA) [47], is used for protein validation. HPA (https://www.proteinatlas.org/) is a manually curated database that collects human proteins in cells, tissues, and organs via integrating various omics technologies such as antibody-based imaging, MS-based proteomics, transcriptomics, and systems biology [47]. Positive proteins for Hela cell line were downloaded (data were downloaded from https://www.proteinatlas.org/search/NOT+celline_category_rna%3AHeLa%3BNot+detected). Proteins having UniProt ids [45] were retained. For Hela cell line, there are 11806 validated proteins (See detail protein list in **S4 Table**). We label the proteins predicted by different protein inference tools, e.g., our ProInfer, EPIFANY, etc., and validated by the Human Protein Atlas as true positives, otherwise they are considered false positives. We calculate several metrics for evaluating competing tools and optimizing their hyperparameters including inferred protein numbers, numbers of true positives, recall, precision and F1 score. Recall, precision and F1 score are given by:

$$recall = TP/(TP + FN) \qquad (16)$$

$$\text{precision} = TP/(TP + FP) \tag{17}$$

$$F1 = 2 \cdot \text{recall} \cdot \text{precision}/(\text{recall} + \text{precision}) \tag{18}$$

where, TP (true positive) refers to a protein reported by an inference tool, e.g., our ProInfer and validated by HPA, FN (false negative) means a protein in HPA but has not been reported by a given protein inference tool, and FP is a protein not in HPA but has been reported.

The final performance evaluation of a tool is determined by the average performance across the 4-replicates' Hela cell line data and varying protein reporting FDR among 0.005, 0.01, 0.025 and 0.05. F1 score is one of the most widely used metrics for measuring performance of a classifier and is used to select optimal hyperparameters.

### Downstream differential expression analysis

A simple differential expression analysis workflow is shown in **Fig 1A**. This was used to benchmark protein inference tools by evaluating their ability to identify differentially expressed proteins. Taking the lung cancer data as an example, proteins differentially expressed in patient samples (4 biological replicates) when compared against normal samples, are expected to be identified. In each of the 8 samples, proteins are inferred by different tools with their matched spectra numbers being counted. An expression matrix for this lung cancer data is formed by integrating the 8 samples' protein (final protein list is a union of all 8 samples) spectra counts where missing proteins are filled with counts of 0. We used this expression matrix as input to edgeR, a widely used differential expression analysis tool [48], to identify differentially expressed proteins. Those proteins with less than 2 non-missing values in samples of each condition are dropped. We define a differentially expressed protein (DEP) as the protein with absolute log2FC $\geq$ 0.585 (FC means fold change, equals to |FC| $\geq$ 1.5) and Benjamini & Hochberg adjusted p-value (adj.pvalue) $\leq$ 0.05 [49].

## Results

### Summary of optimized hyperparameters with Hela cell line data

We identified the optimal running conditions (or settings) for each tool (ProInfer, EPIFANY, Fido, Percolator and PIA) given data of a particular nature. This would allow us to compare the best outcomes possible for each tool.

For each method, during hyperparameter optimization, we ranked their hyperparameters (or combinations of hyperparameters if more than one hyperparameter needs to be tuned) by corresponding average F1 scores across the 4-replicates Hela cell line data. And returned the best hyperparameter/combination.

**Fig 1C and 1D** shows the hyperparameter optimization results of ProInfer. Only PSM filtering threshold (PEP) needs to be selected for ProInfer. Ostensibly, ProInfer has good resistance to low reliability PSMs: When a loose PSM filtering threshold is used, ProInfer achieves higher recall and F1 score with small decrease in precision. For example, when we set PSM filtering threshold as PEP $\leq$ 0.999, we increase coverage by ~400 more correct proteins in Hela cell line than filtering with PEP $\leq$ 0.01 (**Fig 1C**, 4595–4204). However, this comes at the cost of introducing ~300 more false positive proteins (**Fig 1C**, 4806–4512). When filtering with PSM PEP $\leq$ 0.25, the highest precision of 0.963 is obtained, which is 0.007 bigger than PEP $\leq$ 0.999 (0.956), but 0.02 less (**Fig 1D**, 0.554–0.534) for recall and 0.021 less (**Fig 1D**, 0.390–0.369) for F1 score. Notably, stricter filtering condition also eradicated many target peptides, resulting in

the loss of signals that are potentially rescuable via integration with network information. Hence, a looser PSM filtering threshold for ProInfer is preferred. Accordingly, PSM PEP ≤ 0.999 is set as default hyperparameter for ProInfer in following tests.

We tuned hyperparameters for EPIFANY, Fido, Percolator and PIA accordingly. EPIFANY works best when setting PSM PEP ≤ 0.05 and parameter greedy_group_resolution to be "remove_associations_only" and without regularize. For Fido, its optimal hyperparameters are: PSM PEP ≤ 0.1 and greedy_group_resolution setting as "true". Like ProInfer, Percolator also works best with PSM PEP ≤ 0.999. For PIA, inference method of Spectrum Extractor, multiplicative scoring method and PSM PEP score are chosen, and PEP ≤ 0.999 is used. More details are found in **S2 Table**.

## Benchmarking competing tools with THP1 cell line data

We used the technical replicates of M0 THP1 (**Table 1**) for conducting independent benchmarking. Positive proteins were obtained from HPA (11584 proteins, see **S4 Table**, data can be downloaded via https://www.proteinatlas.org/search/NOT+celline_category_rna%3ATHP-1%3BNot+detected). For each tool, optimal hyperparameters were determined as described above. The average performance across the 3 M0 THP1 replicates were used. In addition, an alternative positive protein set generated by filtering out proteins in HPA but without protein level evidence (with 11483 proteins) were also tested, minor performance differences were obtained, see **S7 Table** for more details.

In **Fig 2A**, we showed the proportions of true positives (deep colors) against false positives (light colors). Protein reporting FDRs were set as 0.005, 0.01, 0.025 and 0.05 respectively. Regardless of FDR threshold, ProInfer reports the most numbers of true positives (albeit, with correspondingly more false positives as well). For instance, given FDR 0.01, ProInfer reports 5390 proteins in total, of which, 5041 are true positives. Compared against Percolator, 640 (**Fig 2A**, 5041–4401) more true positives were reported with just 230 (**Fig 2A**, 349–119) more false positives, achieving a 1:2.78 ratio of false positives:true positives gain (230:640). Similarly, the false positives:true positives gains comparing to EPIFANY, Fido and PIA are 1:2.90, 1:6.1 and 1:9.37. In addition, from **Fig 2A**, even with looser FDR thresholds, e.g., 0.025 and 0.05, ProInfer reports more true positives without incurring great changes to the presence of false positives (about 10 more false positives comparing to FDR 0.005 or 0.01). From **Fig 2B**, ProInfer produces stable precisions while other tools acquire lower precisions as FDRs relaxes. In **Fig 2C and 2D**, the line plots show that ProInfer always achieves the highest recall and F1 score. Notably, all methods obtain better recalls and F1 scores when FDR relaxes from 0.005 to 0.05. Amongst the methods, Percolator always gets highest precisions but lowest recalls and F1 scores.

In **Fig 2E**, we also used an upset plot to investigate overlaps among different methods' reported proteins at FDR 0.01 in an example replicate of THP1 M0 cell line (refers to M0_1). We identified 4093 proteins commonly reported by all 5 tools. The overlap amongst competing tools is deep, making up at least 75% of total reported proteins (from 77% for ProInfer to 92% for Percolator). Almost all EPIFANY reported proteins are also reported by at least one tool. Notably, each of the remaining four tools can identify some proteins missed by others. Thus, we added an additional Venn diagram on top of the upset plot to show the reliability of these tool-unique proteins (**Fig 2E** inset). Amongst the 4093 commonly reported proteins, 4010 (98%) is validated by Human Protein Atlas (HPA detectable). Importantly, ProInfer identified the biggest number (1014) of uniquely reported proteins of which 776 out of 1014 (76.5%) were validated. PIA uniquely reported just 166 proteins, with 57.8% (96 out of 166) validated. Fido uniquely reported 83 proteins with less than half (38 out of 83) validated. For

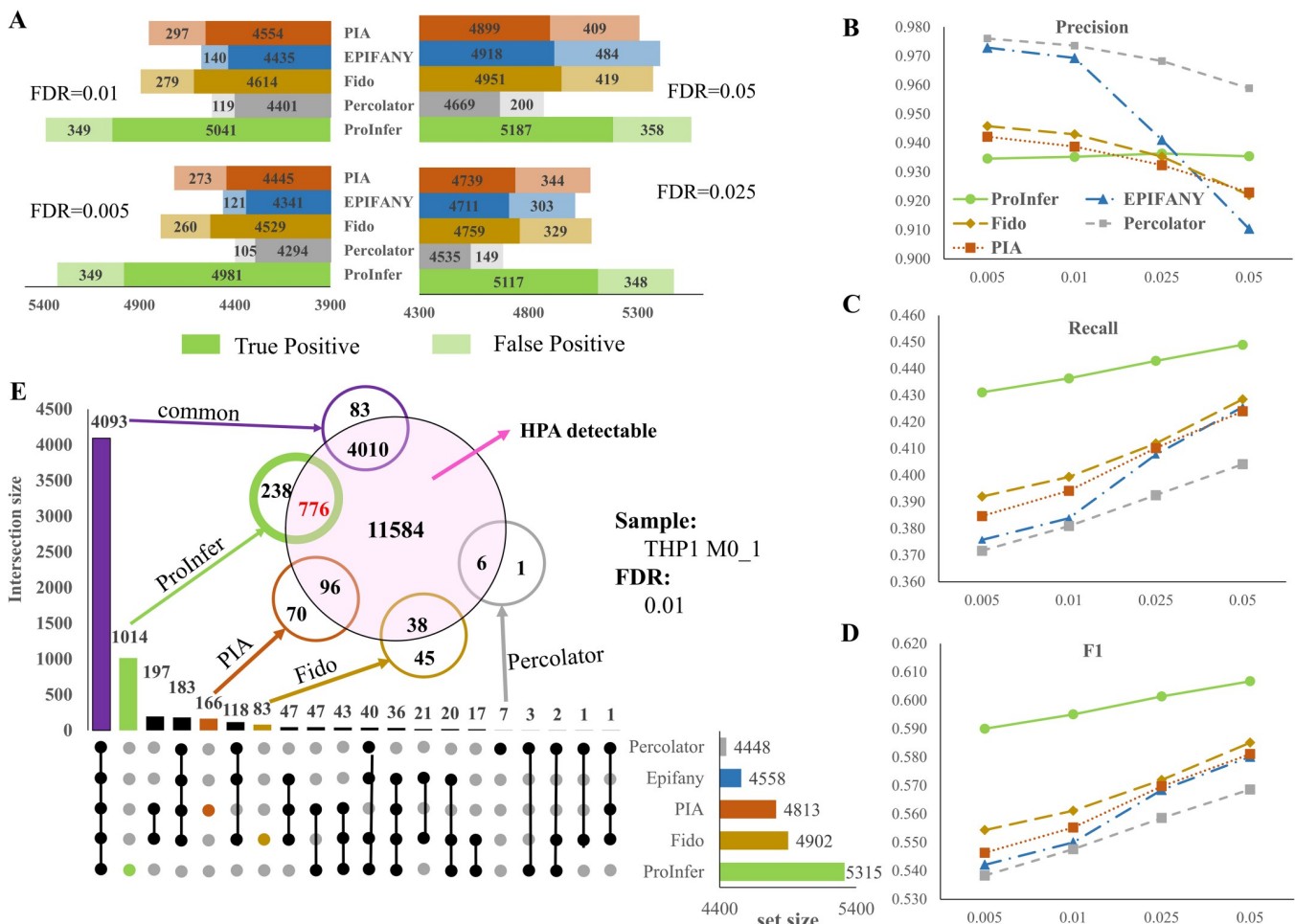

**Fig 2. Performance evaluation of various tools tested on 3 replicates of human THP1 dataset. A** shows average numbers of true positives (in deep colors) and false positives (in light colors) reported in 3 replicates of THP1 sample with FDR of 0.005, 0.01, 0.025 and 0.05 respectively with hyperparameters optimized with Hela data. **B, C & D** show the changes of precision, recall and F1 values of competing tools under different protein reporting FDRs. **E** is a Venn diagram revealing the overlap of proteins reported by different tools and validation status of proteins that reported only by a specific tool.

Percolator, the validation rate of its uniquely reported proteins is 85.7% (6/7), however the list size is very small in comparison. Hence, the superior performance of ProInfer to report many reliable proteins, not found by any other tool, makes it a promising tool to find novel protein biomarkers.

## Evaluations of protein inference tools in differential expression analysis

Following protein inference, downstream quantitative analysis is a key step for proteomics data analysis. To investigate how protein inference tools affect differential expression analysis, we used 3 published datasets for testing. FragPipe [44] and maxquant [50] are two popular platforms for proteome quantification, thus are tested for comparison.

Bar plots in **Fig 3A–3C** present found DEP (in light colors) and non-DEP (in deep colors) numbers in human lung cancer, mouse RAW264.7 and human THP1 data by five protein inference tools, FragPipe and maxquant-based workflows. Given lung cancer and THP1 data, ProInfer reports the most proteins (including DEPs and Non-DEPs) comparing to other four protein inference tools and FragPipe or maxquant. In RAW264.7 data, PIA inferred the most

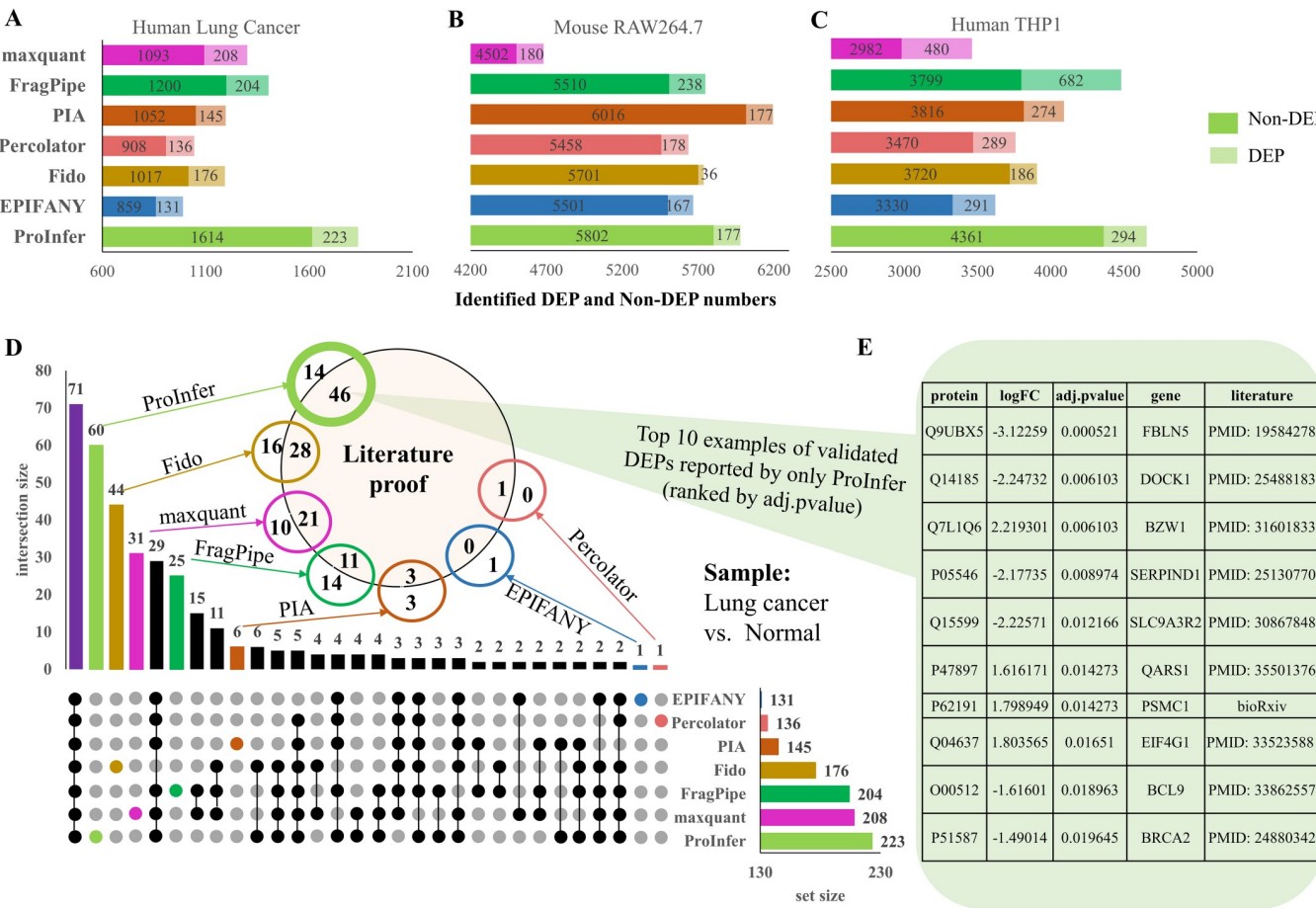

**Fig 3. Evaluation of methods on differential expression analysis.** A, B & C show the numbers of differentially expressed proteins (DEPs) and non-differentially expressed proteins (Non-DEPs) found by different protein inference tool-based, FragPipe and maxquant-based workflows from lung cancer, RAW264.7 and THP1 cell line data. D shows overlapping of DEPs reported by different workflows from lung cancer data and the validation status of those uniquely found DEPs. E gives the top10 example validation proofs of DEPs uniquely found by ProInfer.

proteins compared to other tools, however, ProInfer still found more proteins than other workflows except for PIA. For DEPs, FragPipe and maxquant always report more DEPs than these five protein inference tools except for the lung cancer data, where ProInfer identified about 20 more DEPs. This may be due to both FragPipe and maxquant adopting the match-between-runs (MBR) method to mitigate the missing value problem [51], where smaller missing rates will be obtained, e.g., about 15% for both FragPipe (15.11%) and maxquant (15.41%), but more than 19% for ProInfer (19.43%) and PIA (22.77%) averagely in THP1 M0 and M1 samples. ProInfer performs stably in protein reporting and DEP identification compared to other methods given the three tests.

We are interested in how DEPs identified by different workflows overlap. In **Fig 3D**, an upset plot displays the intersections of DEPs found in lung cancer data by 7 workflows. 71 DEPs were reported by all 7 workflows, comprising 54% of all DEPs found by EPIFANY-based workflow (highest) and 32% of ProInfer-based workflow (lowest). All 7 workflows reported some unique DEPs, which is interesting. These unique DEPs may be valuable, e.g., novel biomarkers or drug targets. We drew an additional Venn diagram (**Fig 3D** inset) to check the validation status of these unique DEPs, where literature proofs were searched to prove they really associate with lung cancer. We searched literature by key words of given

protein and lung cancer at first. Then we read the literature to check whether they reported any correlations between the protein and lung cancer. ProInfer found the largest number of uniquely reported DEPs, where 46 out of 60 of these DEPs can be validated by literature. Similarly, 44 DEPs were uniquely reported by Fido and 28 of them can be proved to play significant roles in lung cancer. For FragPipe and maxquant, 31 and 25 unique DEPs were confirmed and more than a half of them are supported by literature. In comparison, EPIFANY, PIA and percolator reported few unique DEPs. We listed the top 10 DEPs unique to ProInfer ranked by their adj.pvalues in **Fig 3E**. We can find published papers or recent preprints to confirm that these DEPs are lung cancer related. For example, the protein Fibulin-5 (Uniport id: Q9UBX5) is a product of gene FBLN5, which was reported to suppress lung cancer invasion by inhibiting matrix metalloproteinase-7 expression [52]. Pan et al. found that Dedicator of cytokinesis protein 1 (Uniport id: Q14185, product of gene DOCK1) plays significant role in cell migration, Akt expression, and vimentin phosphorylation and it's a drug target for lung cancer [53]. However, these two important DEPs were missed by all other tools. More details about the literature proofs for these uniquely reported DEPs can be found in **S3 Table**. In addition, those unique DEPs with no existing evidence may be novel lung cancer related proteins. These examples show that ProInfer improves protein inference and DEP identification. It is thus useful for biomarker or drug target identification.

## Discussion

### Most protein assembly methods have limited coverage of underlying proteomes

In HPA, there are 11584 confirmed proteins in THP1. Most protein inference methods except ProInfer, identified fewer than 5000 true positives even at a loose protein FDR of 0.05 (**Fig 2A**). Percolator has the worst proteome coverage. This may be due to elimination of ambiguous peptides alongside a simple approach towards protein inference. Though, both ProInfer and other protein inference tools such as Percolator apply a loose PSM filtering threshold, i.e., $PEP \leq 0.999$, ProInfer always achieves excellent performance in protein inference, where highest recalls and F1 scores are always obtained. This may means dropping low confidence peptides (including ambiguous peptides) too early adversely impact proteome coverage. Using biological networks, ProInfer is a successful method that can make good use of peptides with weaker signals (or with lower confidence that may be dropped by stricter filtering conditions) to achieve good proteome coverage.

### While ProInfer dominates in our benchmarks, it does have drawbacks

ProInfer works well on protein inference, especially when a looser peptide filtering criterion, e.g., PSM $PEP \leq 0.999$, was applied. A strict filtering criterion may stave off more decoy peptides, which benefits some tools e.g., EPIFANY and Fido (see **Results**). However, such conservatism also results in widespread loss of informative target peptides, reducing proteome coverage. To manage noise from looser criteria, ProInfer integrates biological network information (e.g., protein complexes) to make good use of those peptides possessing relatively lower confidences to rescue more target proteins. In our evaluations, this strategy has proven effective in identifying more true positives than existing tools.

However, ProInfer has several drawbacks:

Firstly, ProInfer cannot manifest its full potential should biological networks be inapplicable or unavailable in the analytical context (e.g., when we don't have enough reliable protein complexes or there is no complex network formable in the sample). In the current version of

CORUM 3.0, there are 2916 curated human protein complexes, which is quite small, and does not account for all possible networks and complexes partook by all human proteins. The lack of a tissue-specific complexome database is a further limiting factor that we hope can be overcome eventually. Moreover, there are limited number of other species with well-characterized and extensive protein complex lists in this database; thus, unless homology mapping is an option, ProInfer may not work well on samples from other species (e.g., mouse, where PIA reports more proteins from mouse RAW264.7 cell line data).

Secondly, we see potential for further optimization: The parameters may yet be further explored for ProInfer. For example, previously in a related study, we only used the complexes with size $\geq 5$ to reduce instability issues [25]. Here, all curated complexes were used; if a threshold of 5 were used, many complexes would be unavailable, resulting in loss of many weak signal proteins. In future optimizations, we may study the impact of protein complex filtering on ProInfer performance. Moreover, during the calculation of protein complex existence probabilities, only a subset of proteins in a complex and in the candidate protein list ($cP^j \cap Pros$) are considered. The low coverages of complexes may cause overestimation of the probabilities of complexes being present. However, if we filter out these low coverage proteins, then only a few complexes are usable, thus the performance of ProInfer also decreases. Other settings to be tuned includes how to better determine the probability of a complex to be present from its proteins and the signal propagation approach from complexes back to other same-complex proteins. Here, we simply assume the probability of a complex to be present equals to the **maximum** posterior probability of its proteins while the probability of a protein to be present from the complex side is measured as the **maximum** probability of all complexes containing it. In **S5 Table**, we tested setting a complex's probability to be present as the **mean** posterior probability of its proteins. However smaller F1 scores are always achieved especially when a higher psm threshold is configured. Though using mean helps reduce false positive rates (within 1%), much more true positives are also dropped (~10%). Using maximum is currently an optimal selection, more advanced methods that help reduce false positives but keep true positives could be tried, e.g., calculate a prior probability of a complex to be present with enrichment test or our previous weighted probability method [25].

Thirdly, for paralogous proteins that share the same peptides and can participate as mutually exclusive partners in protein complexes [54], ProInfer may possibly infer them as either simultaneously present or absent. This is because ProInfer is dependent on prior knowledge captured in the protein complex databases. It is possible to extend ProInfer by enriching protein complex data with information on gene expression and paralogs. This may reduce potential false positives.

## Future work

Our future work will focus on three aspects. Firstly, we may also try to incorporate tissue-specific expression gene information, e.g., from database TissGDB [55] and housekeeping gene information, e.g., from HRT Atlas [56], to help the identification of proteins with higher confidence of existence based on their biological functions. Such proteins can be accorded higher confidence scores even if their observable peptides present with low signals. Secondly, to cater for big data, we may implement ProInfer in more efficient programming languages, e.g., Scala [57] or C [58]. Finally, while we have evaluated across selective yet high-quality data, there are many new technological advances. Hence, we may further evaluate ProInfer on exciting new data such as single-cell proteomics [59] and spatial proteomics [60].

## Conclusion

We propose a novel biological network-guided method ProInfer for performing inference. ProInfer maximizes use of peptide information (including ambiguous peptides) via a simple yet logical assignment rule. More importantly, biological networks, in the form of protein complexes, is integrated with ProInfer to rescue proteins with weak signals. In our evaluations, ProInfer is robust, and stable even across a wide range of conditions. This is in stark contrast to most other protein assembly tools which are sensitive to adjustments of filtering parameters (especially important since the optimal cutoff is often unknown). Critically, ProInfer can identify large numbers of validated novel proteins not found by any other tool. In our evaluations, we find that these novel proteins are phenotype relevant. Thus, ProInfer is promising for functional profiling and discovering novel biomarkers or drug targets. Source codes of ProInfer are publicly accessible at https://github.com/PennHui2016/ProInfer.

## Supporting information

**S1 Table. Data for generating figures in the main text.** Sheets in S1_Table.xlsx with names "Fig 1C" and "Fig 1D" show the tables containing the data for generating **Fig 1C** and **Fig 1D** in our main text. Similarly, sheets "Fig 2A" to "Fig 2E" and "Fig 3A" to "Fig 3E" show the corresponding data for generating our **Fig 2A** to **Fig 2E** and **Fig 3A** to **Fig 3E** in main text respectively.
(XLSX)

**S2 Table. Results of parameter optimization for competing protein inference tools.** The five Sheets with names "supp.tab1" to "supp.tab5" show the parameter optimization results for EPIFANY, Fido, Percolator, PIA and ProInfer.
(XLSX)

**S3 Table. Literature proofs for validating uniquely found differentially expressed proteins based on different protein inference tools.** The seven sheets with names "supp.tab1" to "supp.tab7" show the literature proofs for validating uniquely found differentially expressed proteins based on five protein inference tools ProInfer, EPIFANY, Fido, Percolator, PIA and two quantification analysis platforms FragPipe and maxquant.
(XLSX)

**S4 Table. Proteins obtained from the Human Protein Atlas for protein inference validation.** Sheets "Hela detectable Protein in HPA" and "THP1 detectable Protein in HPA" contain the lists of detectable proteins in Hela and THP1 from the Human Protein Atlas for validating protein inference performances.
(XLSX)

**S5 Table. Comparison of methods "mean" and "max" for calculating protein complex confidence scores.** Sheet 1 shows the comparison results of using "mean" and "max" to calculate protein complex confidence scores.
(XLSX)

**S6 Table. Results of testing the robustness of proposed ProInfer by removing pre-inferred proteins.** Sheet 1 shows the testing of robustness of ProInfer by removing 5~50% of pre-inferred proteins.
(XLSX)

**S7 Table. The alternative validation data and the comparisons of validating inferred proteins with original validation data and the alternative validation data.** Sheets

"HPA_Hela_protein_level" and "HPA_THP1_protein_level" give the two alternative validation data by removing proteins without protein level evidence. Sheet "parameter optimization" shows the detail parameter optimization results of different protein inference tools based on the alternative validation data in sheet "HPA_Hela_protein_level". The Sheets "Fig 1C", and "Fig 1D" show the minor changes in result data for generating our **Fig 1C** and **Fig 1D** in main text when using the alternative validation data in sheet "HPA_Hela_protein_level". Sheets with names "Fig 2A" to "Fig 2E" show the minor changes when using the alternative validation data in sheet "HPA_THP1_protein_level" for performance tests of different protein inference tools.
(XLSX)

## Author Contributions

**Conceptualization:** Hui Peng.

**Data curation:** Hui Peng.

**Formal analysis:** Hui Peng.

**Investigation:** Hui Peng.

**Methodology:** Hui Peng.

**Supervision:** Limsoon Wong, Wilson Wen Bin Goh.

**Writing – original draft:** Hui Peng.

**Writing – review & editing:** Hui Peng, Limsoon Wong, Wilson Wen Bin Goh.

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
