## [Decision Letter · Decision Letter 0]

15 Nov 2022

Dear Dr. Peng

Thank you very much for submitting your manuscript "ProInfer: An interpretable protein inference tool leveraging on biological networks" for consideration at PLOS Computational Biology.

As with all papers reviewed by the journal, your manuscript was reviewed by members of the editorial board and by several independent reviewers. In light of the reviews (below this email), we would like to invite the resubmission of a significantly-revised version that takes into account the reviewers' comments.

In particular I would suggest to focus on the issue of false positives raised by the reviewers and the performance comparison with other methods available. In addition is of paramount importance that your tools are accessible and usable without errors, therefore your github repository needs to be updated and kept up to date and flawless.

We cannot make any decision about publication until we have seen the revised manuscript and your response to the reviewers' comments. Your revised manuscript is also likely to be sent to reviewers for further evaluation.

Sincerely,

Franca Fraternali

Guest Editor

PLOS Computational Biology

Lucy Houghton

Staff

PLOS Computational Biology

Reviewer's Responses to Questions

**Comments to the Authors:**

Reviewer #1: The authors developed a protein inference tool named ProInfer. The proposed method makes use of prior knowledge from biological networks to improve the performance of protein assembly based on MS data. ProInfer showed superior performance compared to several other publically available tools. The tool was written in Python, and the source code is openly available. The work is of value, and ProInfer can be a valuable tool for proteomic data analysis.

To make the tool more accessible, especially to colleagues without computational background, I suggest, in addition to releasing the source code, providing a stand-alone version or a web-based application.

Reviewer #2: ProInfer is a computational analysis framework which infers the identification of proteins from ambiguous peptides identified in mass spectrometry data. The conceptual advance ProInfer provides is incorporating information from protein interaction networks such as CORUM (i.e. protein complexes) to improve the identification of proteins in a given sample. The authors’ compare their method against other protein inference algorithms on a benchmark of Human Protein Atlas validated/unvalidated proteins and show superior recall although limited relative precision. The authors’ also include an analysis of differentially expressed proteins and compare their results to other popular algorithms showing mostly positive performance.

Major:

1) Integration of protein complex information to improve protein identification may also introduce bias in the true positives and false positives proteins identified by the method. The authors’ alluded to some of the reasons in their discussion for true positive bias, in particular the incompleteness of biological networks. This however is not the only reason for bias, and I feel a full analysis of the false positives is necessary to give a complete picture.

One possible source of false positives is partners of proteins that participate in many complexes. It is unclear how these cases are handled in the ProInfer framework. This is a major concern due to the use of maximum probability values. Calmodulin (CALM1), for example, is found in 11 different Human Corum complexes many of them with non-overlapping subunits and unlikely to be expressed all together. If I understand the algorithm correctly, in a scenario where CALM1 is highly (or moderately) abundant in a sample, step 2 will assume all 11 CALM1 containing complexes are present regardless of if the complexes are actually formed, expressed, or seen previously in that cell type/condition. Further in Step 4, using the maximum will transfer the probability of the most confident protein (CALM1 in this case) to all other interaction partners in the 11 complexes regardless of if the individual subunits are expressed or not. This potentially is one source of false positives the method is producing.

An additional issue that is unclear is how the framework handles paralogous proteins. Paralogous proteins often share the same peptides and can participate as mutually exclusive partners in protein complexes. This method suggests if a partner in a shared complex is present, both paralogous proteins would be identified.

Due to the above issues and possible other unknown biases, a more complete analysis of the false positives generated by the method is needed and a description of the biases introduced into the identified proteins due to the use of protein complexes as a source of integrated information.

2) It is unclear how the decoy protein complexes are utilized in the method. It is mentioned but not fully described.

3) The Human Protein Atlas is used as a benchmark. It is unclear which dataset within the HPA was used. A link to the downloads page would aid in replicability. Further, the description of “validated” vs “non-validated” is vague. What do the authors’ mean by “different tools”? Is it only antibody based methods? Specifics are required for a full evaluation of the method. Publishing the full benchmark in the supplemental along with ProInfer’s identified proteins is necessary as well.

4) The text states “… literature proofs were searched to prove they really associate with lung cancer.” It is unclear what the criterion is for inclusion here. Many studies have extremely weak associations between genes and diseases and rigorous review of the literature is required to ensure strong evidence of associations. The authors should detail their method and include a supplemental table for the reader to evaluate.

5) A fresh clone of the github repository does not produce executable code. I received an error when trying the example command: python ProInfer.py ./DDA1.tsv

NameError: name 'argparse' is not defined

When I added ‘import argparse’ to the file, I received a different error:

ProInfer.py: error: unrecognized arguments: ./DDA1.tsv

Minor:

Some of the axes in the figures are not labeled properly. For example in Figure 3A-C, the x-axis requires a label.

Reviewer #3: In this article, the authors present ProInfer, a method that performs protein identification by using an inference method harnessing protein complex information.

The methods are interesting, and well explained, and the results appear convincing. I have a few comments that I believe should be addressed:

- the independence assumption in equation 6 seems entirely unjustified, especially if overlapping peptides are considered. Can the authors test that assumption? This directly relates to the next point, since this formula is used for the computation of the FDR:

- The flat precision curve for ProInfer in figure 2B is puzzling: the precision should go down as the FDR increase. How do the author explain this behaviour?

- It seems to me that the "reliable" complexes are implicitely given a prior of 1, and the same weight. But different complexes have widely varying concentrations, and confidence. Are they all given the same prior likelihood ?

- how useful is proinfer when pre-existing identifications of proteins are weaker? for example with less identified proteins in the reference? and how does it affect ProInfer's precision and recall? It seems hard to tell whether ProInfer is efficiently using weak peptide identifications, or using noisy peptide spectra to assign proteins from the network.

- it would be useful to see a comparison between proInfer and some of the other tools as a function of psm threshold: does proInfer do better than other tools at smaller psm thresholds?

- It would be nice to have in supplementary some peptide spectra, to illustrate what type of evidence that proInfer uses to draw its inference, that would otherwise be ignored.

**Have the authors made all data and (if applicable) computational code underlying the findings in their manuscript fully available?**

Reviewer #1: Yes

Reviewer #2: **No: **Code is available but does not run without errors. Full results of the code is not provided in the supplementary tables.

Reviewer #3: Yes

PLOS authors have the option to publish the peer review history of their article (what does this mean?). If published, this will include your full peer review and any attached files.

Reviewer #1: No

Reviewer #2: No

Reviewer #3: **Yes: **Tristan Cragnolini
---

## [Decision Letter · Decision Letter 1]

18 Jan 2023

Dear Dr Peng

Thank you very much for submitting your manuscript "ProInfer: An interpretable protein inference tool leveraging on biological networks" for consideration at PLOS Computational Biology. As with all papers reviewed by the journal, your manuscript was reviewed by members of the editorial board and by several independent reviewers. The reviewers appreciated the attention to an important topic. Based on the reviews, we are likely to accept this manuscript for publication, providing that you modify the manuscript according to the review recommendations.

In particular you should carefully address the requested definition of your approach to defining false positive. Please respond carefully to the suggestions of the reviewer both in the first and second revision,

Looking forward to your revised manuscript,

with my best regards

Franca Fraternali

Sincerely,

Franca Fraternali

Guest Editor

PLOS Computational Biology

Lucy Houghton

Staff

PLOS Computational Biology

Reviewer's Responses to Questions

**Comments to the Authors:**

Reviewer #2: I thank the authors for clarifying some of the points in their manuscript.

1. I do still have a concern about certain proteins found in many complexes. As mentioned before, some complexes may not be expressed even though a member is highly abundant (CALM1 as an example). This may lead the algorithm to improperly “enhance” the confidence score of a protein associated to a complex that is not expressed. This may be a source of false positives. In the authors’ response, they describe a scenario of two proteins P1 and P2 which have small confidence scores which map to two different complexes associated to CALM1, c1 and c2. P1 and P2 are both enhanced due to their association to c1 and c2. If we additionally know that c2 is not expressed in the tissue of study, P2’s confidence score was then erroneously enhanced. In the described algorithm, this scenario is only problematic with proteins that are found in many complexes such as CALM1.

A potential test would be to remove all proteins and their complexes from CORUM seen in > 3 complexes and rerun ProInfer. As a control, remove the same number of proteins/complexes that match the average abundances in the first set. This may show if the many complex proteins are having an effect.

2. The use of the terms “validated” and “non-validated” is confusing as it suggests using additional experimental evidence for validation. Using terms “true-positives” and “false-positives” seems sufficient.

**Have the authors made all data and (if applicable) computational code underlying the findings in their manuscript fully available?**

Reviewer #2: Yes

PLOS authors have the option to publish the peer review history of their article (what does this mean?). If published, this will include your full peer review and any attached files.

Reviewer #2: No

Figure Files:

Data Requirements:

Reproducibility:

References:

---

## [Decision Letter · Decision Letter 2]

20 Feb 2023

Dear Dr Peng,

We are pleased to inform you that your manuscript 'ProInfer: An interpretable protein inference tool leveraging on biological networks' has been provisionally accepted for publication in PLOS Computational Biology.

Best regards,

Franca Fraternali

Guest Editor

PLOS Computational Biology

Lucy Houghton

Staff

PLOS Computational Biology

Reviewer's Responses to Questions

**Comments to the Authors:**

Reviewer #2: The new analysis addresses my concerns regarding false positives introduced by proteins common to multiple complexes. I also feel the revised manuscript is clearer with the additional edits.

**Have the authors made all data and (if applicable) computational code underlying the findings in their manuscript fully available?**

Reviewer #2: None

PLOS authors have the option to publish the peer review history of their article (what does this mean?). If published, this will include your full peer review and any attached files.

Reviewer #2: No

---

## [Editor Report · Acceptance letter]

12 Mar 2023

PCOMPBIOL-D-22-01320R2 

ProInfer: An interpretable protein inference tool leveraging on biological networks

Dear Dr Wong,

I am pleased to inform you that your manuscript has been formally accepted for publication in PLOS Computational Biology. Your manuscript is now with our production department and you will be notified of the publication date in due course.

With kind regards,

Zsofia Freund
